# Analysis of Cultivable Microbial Community during Kimchi Fermentation Using MALDI-TOF MS

**DOI:** 10.3390/foods10051068

**Published:** 2021-05-12

**Authors:** Eiseul Kim, Seung-Min Yang, Hae-Yeong Kim

**Affiliations:** Institute of Life Sciences & Resources and Department of Food Science and Biotechnology, Kyung Hee University, Yongin 17104, Korea; eskim89@khu.ac.kr (E.K.); ysm9284@gmail.com (S.-M.Y.)

**Keywords:** MALDI-TOF MS, kimchi, fermented food, lactic acid bacteria, identification, microbial community, biotyper, culture-dependent method, fermented temperature

## Abstract

Kimchi, a traditional Korean fermented vegetable, has received considerable attention for its health-promoting effects. This study analyzes the cultivable microbial community in kimchi fermented at different temperatures using matrix-assisted laser desorption/ionization time-of-flight mass spectrometry (MALDI-TOF MS) to comprehensively understand the factors affecting the quality of kimchi. Of the 5204 strains isolated from kimchi, aligned with the in-house database, 4467 (85.8%) were correctly identified at the species level. The fermentation temperature affected the microbial community by varying the pH and acidity, which was mainly caused by temperature-dependent competition between the different lactic acid bacteria (LAB) species in kimchi. LAB, such as *Levilactobacillus* (*Lb*.) *brevis* and *Lactiplantibacillus* (*Lpb*.) *plantarum* associated with rancidity and tissue softening, proliferated faster at higher temperatures than at low temperature. In addition, LAB, such as *Latilactobacillus* (*Lat*.) *sakei* and *Leuconostoc* (*Leu*.) *mesenteroides*, which produce beneficial substances and flavor, were mainly distributed in kimchi fermented at 4 °C. This study shows as a novelty that MALDI-TOF MS is a robust and economically affordable method for investigating viable microbial communities in kimchi.

## 1. Introduction

Kimchi is known to have health-promoting effects, such as anti-oxidative and anti-aging effects [1,2,3,4]. It is fermented in the unsterilized natural environment, leading to the growth of various microorganisms. The composition of the microbial community changes during fermentation [5]. The fermentation of kimchi is affected by environmental factors, such as fermented temperature and ingredients. During kimchi fermentation, lactic acid bacteria (LAB) are mainly responsible for reducing the growth of aerobic bacteria because of acid production and in creating the flavor profile of kimchi by organic acids, acid production, ethanol, and CO_2_ [6,7,8]. The temperature influences the quality, flavor, and shelf-life of kimchi by affecting the growth of each LAB group [9]. Therefore, it is important to set an accurate temperature to improve the quality, flavor, and taste of kimchi and provide vital information concerning the environmental factors and kimchi ecosystem during fermentation.

Culture-independent approaches are widely used to identify the microbial community in fermented foods [10,11,12]. However, these approaches are time-consuming, expensive, difficult to carry out, labor-intensive, and provide limited information. To overcome these limitations, the high-throughput sequencing approach is used to identify the microbial community in various fermented foods [13,14,15]. It provides a comprehensive understanding of the relationship between food and its microorganisms compared to other culture-independent methods, but its taxonomic resolution is only up to the family or genus level because of the use of 16S rRNA gene fragments [5,13,16]. In addition, it is difficult to observe the viable microbial community because the DNA used in this method is recovered from both live and dead microbial cells [17,18]. Nevertheless, many studies use this method to observe the microbial community in kimchi [5,19].

Matrix-assisted laser desorption/ionization time-of-flight mass spectrometry (MALDI-TOF MS) is a powerful tool for faster and cost-effective identification of microorganisms compared to conventional molecular identification systems. The equipment can be handled easily by anyone to accurately identify even a small number of microorganisms and recover them after the experiment [20,21]. In addition, high-throughput identification is possible due to the rapid handling of a large number of microorganisms [22]. The technique is based on the comparison between the mass spectrum produced by the whole cells and reference spectra, so the reliability of MALDI-TOF MS depends on the quality and quantity of reference mass spectrum profile. Compared to the 16S rRNA gene sequence, which is commonly used for identifying bacterial strains, MALDI-TOF MS obtained similar results. In some cases, it is superiorly capable of taxonomic classification down to species or subspecies level [23,24]. In particular, the role of microorganisms during the kimchi fermentation is species-specific, and therefore, it is critical to accurately analyze the microbial community at the species level. However, the previous studies conducted in this field have only focused on the high-throughput sequencing method, so that microbial community has been identified only at the genus level [1,5,13,25]. Until now, MALDI-TOF MS has been successfully applied to identify various clinical isolates, anaerobes, and fungi [24,26,27]. However, to our knowledge, the application of MALDI-TOF MS to observe the changes in microbial communities in fermented foods is still rare.

The novelty of this study is to investigate the cultivable microbial community in kimchi. The previous studies conducted in this field have only focused on investigating microbial communities based on DNA, in which both live and dead microbial cells are detected [1,5,13,25]. The DNA-based identification methods tend to overestimate bacterial abundance in a sample and fail to reflect the presence of microbial subpopulations with different viability states [28,29]. This study aimed to investigate the changes in cultivable microbial communities in kimchi with different fermentation temperatures using MALDI-TOF MS and to observe the relationship between the microbial communities and the quality of fermented kimchi by confirming the changes in its viable cell count, pH, and acidity.

## 2. Materials and Methods

### 2.1. Sample Preparation and Sampling

The overall procedure for this study is shown in Figure 1. The kimchi samples were prepared according to a standardized method [30]. Briefly, the Chinese cabbages were soaked in 15% saltwater for 10 h and washed with tap water thrice. The salted cabbage was mixed with various ingredients: Red pepper (7.4%), radish (56.7%), green onion (5.3%), garlic (7.9%), ginger (1.1%), sugar (0.7%), seawoo jeotgal (10.5%), and myeolchi aekjeot (10.5%), and then packaged into three plastic containers. Each plastic container with kimchi was stored at the standard fermentation temperature of kimchi (4 °C), sales rack temperature (10 °C), and room temperature (23 °C). The kimchi samples at 4 °C and 10 °C were periodically analyzed from 0 to 60 days at 5-day intervals, while those at 23 °C were periodically analyzed from 0 to 9 days, once a day.

### 2.2. Measurement of Viable Cell Counts, pH, and Acidity

For viable cell counts, 25 g fermented kimchi was added to 225 mL of 0.85% NaCl in sterile filter bags (Seward Limited, London, UK), and the sample was homogenized at 200 rpm for 2 min using a stomacher (Circulator stomacher 400; Seward Limited). Then, 100 µL of homogenized samples were spread onto plate count agar (PCA, Difco, Becton Dickinson, Sparks, MD, USA) plate and lactobacilli MRS agar plate (Difco) after ten-fold dilutions with 0.85% NaCl. The MRS agar plate was incubated anaerobically for 48 h at 30 °C, while the PCA agar plate was incubated aerobically for 48 h at 30 °C before colony counting and identification. To measure the pH and acidity, 50 g of each kimchi sample was ground for 2 min using a blender and was filtered through sterile gauze. The pH of 50 mL filtrate was measured using a pH meter (Orion Star A211; Thermo Fisher Scientific, Waltham, MA, USA) in triplicate. To measure the acidity, 50 mL of filtrate was titrated with 0.1 N NaOH at pH 8.2 in triplicate. The measured volume of 0.1 N NaOH was substituted into the percentage of lactic acid (%, *v*/*v*).

### 2.3. Analysis of Microbial Community in Fermented Kimchi by MALDI-TOF MS

The microbial community in fermented kimchi samples was analyzed by MALDI-TOF MS. In a dilution plate with 100 to 200 colonies, almost all of them were selected for identification. The selected colonies were transferred to a growth agar plate and incubated at 30 °C for 48 h, and the fresh colonies were used for identification. To identify the isolated bacteria, a single colony was directly smeared onto the MSP 96 steel target plate (Bruker Daltonics, Bremen, Germany) and overlaid with 1 µL of 70% formic acid. All spots were dried and overlaid with 1 µL of HCCA (10 mg/mL α-cyano-4-hydroxycinnamic acid) matrix solution in 50% acetonitrile, 47.5% water, 2.5% trifluoroacetic acid. The sample was air-dried until the matrix co-crystallized; then, the target slide was loaded into the Microflex LT bench-top mass spectrometer (Bruker Daltonics). The analysis was performed using FlexControl 3.0 software (Bruker Daltonics) and MALDI bioTyper database (5627 reference spectra) with extended bioTyper database (5633 spectra), as previously published [31]. The automatic measurement method with parameter conditions was as follows: Mass range of 2000 to 20,000 Da, initial laser power at 25%, and maximal laser power at 35%. The calibration of the bacterial test standard was done to the standard calibration mixture using FlexAnalysis 3.4 (Bruker Daltonics). The data were translated from the manufacturer’s standard criteria [32]. As specified by the manufacturer, species level was indicated for log score ≥ 2.0, genera level was indicated for log score < 2.0 and ≥1.7, and not reliable identification was indicated for log score < 1.7.

### 2.4. Statistical Analysis

The viable cell counts, pH, and acidity were determined in triplicates, and the data values are expressed as means ± standard deviations. The relationship between the pH, acidity, temperature, and major species in fermented kimchi was determined by calculating the Pearson correlation implemented in R (version 1.3).

## 3. Results and Discussion

### 3.1. Changes in Viable Cell Count, pH, and Acidity

The viable cell count, pH, and acidity were monitored every sampling time. The changes in pH and acidity during kimchi fermentation are shown in Figure 2. The decrease in pH and increase in acidity changed more slowly at a low temperature (4 °C) than at higher temperatures (10 °C and 23 °C). The pH of 4.2 and 0.6% acidity are considered the optimum values of kimchi with the highest quality [9,33]. At 4 °C, it took about 20 days to reach a pH of 4.2, while it took about 10 days and 2 days for samples at 10 °C and 23 °C, respectively. After 15 days, the pH of kimchi fermented at 4 °C and 10 °C rapidly decreased to 4.61 ± 0.01 and 4.01 ± 0.01, respectively; then, the pH remained stable until the end of fermentation. In kimchi fermented at 23 °C, the pH rapidly decreased to 4.26 ± 0.02 in one day and gradually decreased thereafter. The best value of acidity (0.6%) was reached by the sample at 4 °C for about 20 days. The sample at 10 °C reached the optimum acidity after five days, while the sample at 23 °C reached the optimum after one day. The samples at high temperatures reached the best pH and acidity faster than those at low temperatures. The quality of kimchi is usually unacceptable when the acidity is 1.5–2.0% [9]. In this study, the acidities at 10 °C and 23 °C reached unacceptable levels after 40 and 9 days, respectively, but at 4 °C, the acidities were not reached until after 60 days.

The viable cell counts are shown in Figure 3. The number of viable cells has similar overall patterns depending on the medium. At 4 °C and 10 °C, the number of viable cells reached the maximum in 15 days, with an average of 7.86 ± 0.10 log CFU/mL and 7.94 ± 0.06 log CFU/mL, respectively, and it slightly decreased until the end of fermentation. The kimchi fermented at 23 °C required less time than the samples at 4 °C and 10 °C to reach the maximum total cell counts with an average of 7.85 ± 0.05 log CFU/mL after three days of fermentation. The number of viable cells appeared to be related to pH and acidity, which gradually increased and then decreased throughout the fermentation. As a result of the changes in viable cell counts, pH, and acidity, it was found that the quality was maintained for a long time when kimchi was fermented at low temperatures.

### 3.2. Identification of Kimchi Isolates with an In-House Database

In a previous study, the in-house database was constructed to improve the identification rate of strains isolated from traditional Korean fermented foods [31]. Before identifying the isolates, eight reference strains mainly involved in kimchi fermentation were analyzed using an in-house database, and it was able to accurately identify the reference strains (Figure 4). After confirming this, the spectra of 5204 kimchi isolates were aligned with the in-house database. Many isolates were identified exactly, of which 4467 isolates (85.8%) were identified at the species level (log score ≥ 2.0), and 737 isolates (14.2%) were identified at the genus level (log score < 2.0 and ≥1.7) (Table 1). Therefore, it was confirmed that the MALDI-TOF MS method used in this study can sufficiently identify the isolated bacteria related to kimchi fermentation.

### 3.3. Changes in Microbial Community during Kimchi Fermentation

Previous studies used high-throughput sequencing targeting 16S rRNA gene sequences to observe the microbial community in fermented foods [5,13]. However, it provides low taxonomic resolution for some closely related species. In particular, bacteria, such as *Lactiplantibacillus* (*Lbp.*) *plantarum* group (99.4–99.9% similarity), *Latilactobacillus* (*Lat.*) *sakei* group (99% similarity), and *Weissella* (*W*.) *cibaria/W. confusa* (99.2% similarity), which are mainly involved in kimchi fermentation, were not accurately distinguished at the species level [13,31,34]. In contrast, these species were correctly distinguished by MALDI-TOF MS, assuming that the database was sufficient [31]. In addition, the ribosomal protein provides greater identification than 16S rRNA gene sequences, thus, the ribosomal proteins can be used as an alternative to the 16S rRNA gene. Therefore, in this study, the microbial community in kimchi that changed during fermentation at different temperatures were investigated at the species level using MALDI-TOF MS.

A total of 5204 strains isolated from fermented kimchi were identified using MALDI-TOF MS, and the microbial community is shown in Figure 5. *Lactobacillus*-related genera, *Weissella*, and *Leuconostoc* were the dominant genera during kimchi fermentation, which was consistent with the results of previous studies [5,13]. The composition of these microorganisms was similar to sauerkraut, a fermented cabbage food, but it was different from other fermented soybean foods, such as gochujang [15,35]. The overall change in microbial diversity based on fermentation temperature-time was similar to fermentation at 10 °C and 23 °C, but it was different at 4 °C. Immediately after making kimchi, many species, such as *W. cibaria* (27.3%), *Lactococcus* (*Lc*.) *garvieae* (20.7%), *Lc. lactis* (20.0%), *Levilactobacillus* (*Lb*.) *brevis* (19.3%), and *Lpb. plantarum* (11.3%), were identified. In kimchi fermented at 4 °C, *W. cibaria* dominated the early stage of kimchi fermentation, but it disappeared rapidly after 10 days when *Lat. sakei* increased. *Lat. sakei* increased in the middle of fermentation, and it was replaced by *Leu*. *mesenteroides* in the late stage. At 10 °C and 23 °C, *Lb. brevis* and *Lpb. plantarum* remained relatively stable until the end and became the dominant species in the LAB community, with an abundance that reached 100%.

In previous studies that analyzed the microbial community in kimchi using high-throughput sequencing targeting 16S rRNA gene sequences, a similar microbial composition was observed in this study [1,5]. *Lactobacillus*-related genera, *Leuconostoc*, and *Weissella* were mainly identified in various kimchi samples by high-throughput sequencing method, and *Lat. sakei* group, *Leu*. *mesenteroides*, and *W*. *cibaria* were also found at the species level [1,5,13]. However, since the high-throughput sequencing method cannot accurately identify the kimchi microorganisms, such as *Lat. sakei*/*Lat. curvatus* and *W. cibaria*/*W. confusa* with very similar 16S rRNA gene sequences, MALDI-TOF MS may provide higher resolution [13].

### 3.4. Relationships between Microbial Composition and Environmental Factors

The correlation between the relative abundance of major species and environmental factors, such as pH, acidity, and fermented temperature, was analyzed to determine whether these affect the microbial community (Figure 6). *W. cibaria* was relatively abundant immediately after making the kimchi, and it was presumed that they are derived from ingredients, such as salted cabbage and green onion. *W. cibaria* was present at the beginning of fermentation at all temperatures, but it decreased with the increase in *Lactobacillus*-related species as the fermentation progressed. This species had a positive correlation with pH (Pearson coefficient *r* = 0.636, *p* = 0.0001), with a tendency to decrease when pH was lowered. This finding corresponds to a previous study, where the proportion of *Weissella* decreased in the middle of fermentation because the acidity affected its growth [36].

*Lactobacillus*-related species, especially *Lat. sakei*, *Lb. brevis*, and *Lpb. plantarum*, were detected in fermented kimchi samples. *Lat. sakei* grow well and is mainly involved in the fermentation of vacuum-packed meat products at low temperatures [37]. It also dominated the fermentation of kimchi at 4 °C and appeared during the middle until the end of fermentation. This species was also identified with low abundance in the early stage of kimchi fermentation at 10 °C, but not at 23 °C. In the relationship between the bacterial composition and environmental factors, *Lat. sakei* is negatively correlated with fermented temperature (Pearson coefficient *r* = −0.523, *p* = 0.003) (Figure 6).

In previous studies, *Lb. brevis* and *Lpb. plantarum* were isolated in the later stage of fermentation and reported to cause rancidity in fermented kimchi in the middle to end stage at high temperatures [13]. In this study, *Lbp. plantarum* showed a pattern similar to *Lb. brevis* and was found at all fermentation temperatures. However, unlike *Lat. sakei*, it was dominant only at high temperatures (10 °C and 23 °C). In the relationship between bacterial composition and environmental factors, *Lb. brevis* (Pearson coefficient *r* = −0.647, *p* = 8.363 × 10^−5^) and *Lpb. plantarum* (Pearson coefficient *r* = −0.607, *p* = 0.0003) are negatively correlated with pH value. On the contrary, acidity is positively correlated with *Lb. brevis* (Pearson coefficient *r* = 0.749, *p* = 1.254 × 10^−6^) and *Lpb. plantarum* (Pearson coefficient *r* = 0.905, *p* = 2.813 × 10^−12^). Its relationship with fermentation temperature showed a weak positive correlation, and similar to previous studies, these species adapted well to high temperatures and acidic environments.

*Leu. mesenteroides* grow at temperatures below 10 °C and is often isolated from meat products stored at low temperatures [38]. In this study, *Leu. mesenteroides* was present from the middle to the end of fermentation at 4 °C and did not exist in fermented kimchi at 10 °C and 23 °C. Correlation between the abundance of species and the fermented temperature was analyzed, and the result indicated that *Leu. mesenteroides* was species having a negative correlation with fermented temperature (Pearson coefficient *r* = −0.432, *p* = 0.015). Therefore, this result suggests that kimchi should be fermented at low temperatures to increase the proliferation of *Leu. mesenteroides*, a beneficial bacterium that provides the flavor of kimchi [30].

The correlation analysis between the bacterial species and environmental factors showed that *W. cibaria* did not adapt well in an acidic environment, whereas some *Lactobacillus*-related species, such as *Lpb. plantarum* and *Lb. brevis*, adapted well. It was also confirmed that *Lat. sakei* and *Leu. mesenteroides* grew well at low temperature, while *Lpb. plantarum* and *Lb. brevis* grew well at high temperatures. In conclusion, the dominant LAB species are different depending on the fermentation temperature, and this can affect the flavor and fermentation of kimchi. This suggests that fermentation temperature is an important index that determines the quality of kimchi.

### 3.5. Greenness Assessment

The green analytical procedure index (GAPI) is a tool to evaluate the green character and the environmental impact of the entire analytical methodology, from the collection of samples to the final determination [39]. Evaluation of greenness of the proposed MALDI-TOF MS method was performed according to the GAPI tool [39]. Appendix A demonstrates the GAPI pictograms organized according to data collected in Appendix A. GAPI uses five pentagrams to assess environmental impacts at each step of the methodology with three color codes; green, yellow, and red representing low, medium, and high effects on the environment, respectively [40]. According to Appendix A, the GAPI pictogram for the proposed method has no red code with almost green zones except for the four yellow codes (middle impact), which correspond mainly to the sample extraction and safety of the reagents used. Therefore, the result from the pictogram indicates that the proposed method has eco-friendly, due to the low consumption of reagents and solvents and small waste production in MALDI-TOF MS.

## 4. Conclusions

In this study, MALDI-TOF MS was successfully applied to monitor the microbial community in kimchi fermented at different temperatures. Based on MALDI-TOF MS, kimchi microorganism could be identified up to the species level, and only living cells were included so that the dominant species could be identified accurately. The most dominant bacterial species in fermented kimchi were *W. cibaria*, *Lat. sakei*, *Lpb. plantarum*, *Leu. mesenteroides,* and *Lb. brevis*. The relative abundance of LAB in fermented kimchi was related to the changes in fermentation temperature, pH, and acidity. The MALDI-TOF MS approach used in this study can be used to observe the microbial changes and microbial clusters of fermented food and environment. The microorganisms obtained in this experiment can be used for further studies, such as functional LAB studies and starter culture development. The future direction is recommended to gain a deeper understanding of the microbial population by investigating the non-viable microbial communities in kimchi fermentation.

## Figures and Tables

**Figure 1 foods-10-01068-f001:**
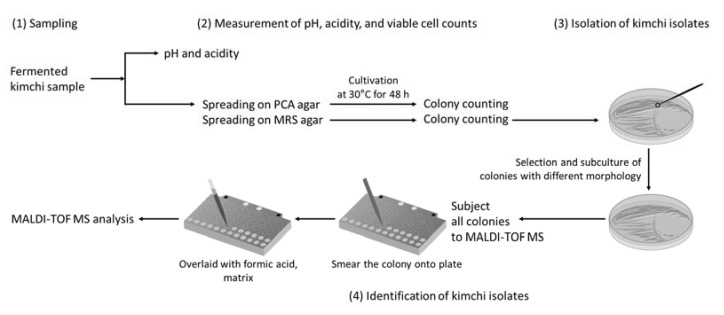
Diagram for analyzing the cultivable microbial community in kimchi. The strategy to identify microbial community involves: (1) Sampling, (2) Measurement of and viable cell counts, pH, and acidity, (3) Isolation of kimchi isolates, and (4) Identification of kimchi isolates.

**Figure 2 foods-10-01068-f002:**
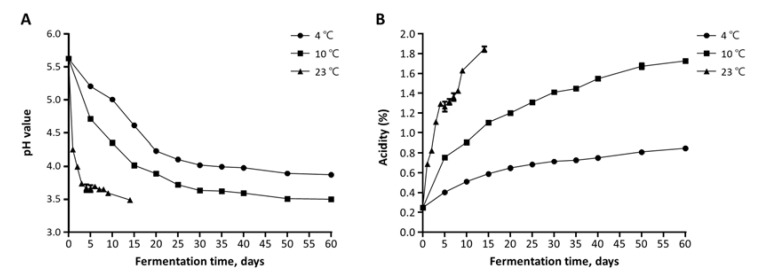
Changes in (**A**) pH and (**B**) acidity of kimchi samples during the fermentation at different fermented temperatures. Error bars represent the standard deviation among the three replicates.

**Figure 3 foods-10-01068-f003:**
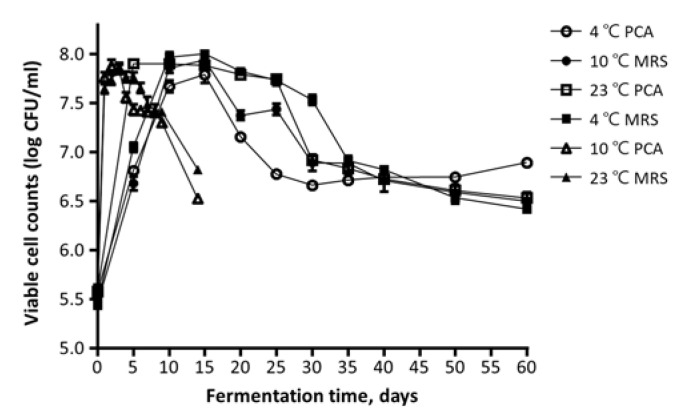
Changes in viable cell count of kimchi samples during the fermentation period at different temperatures.

**Figure 4 foods-10-01068-f004:**
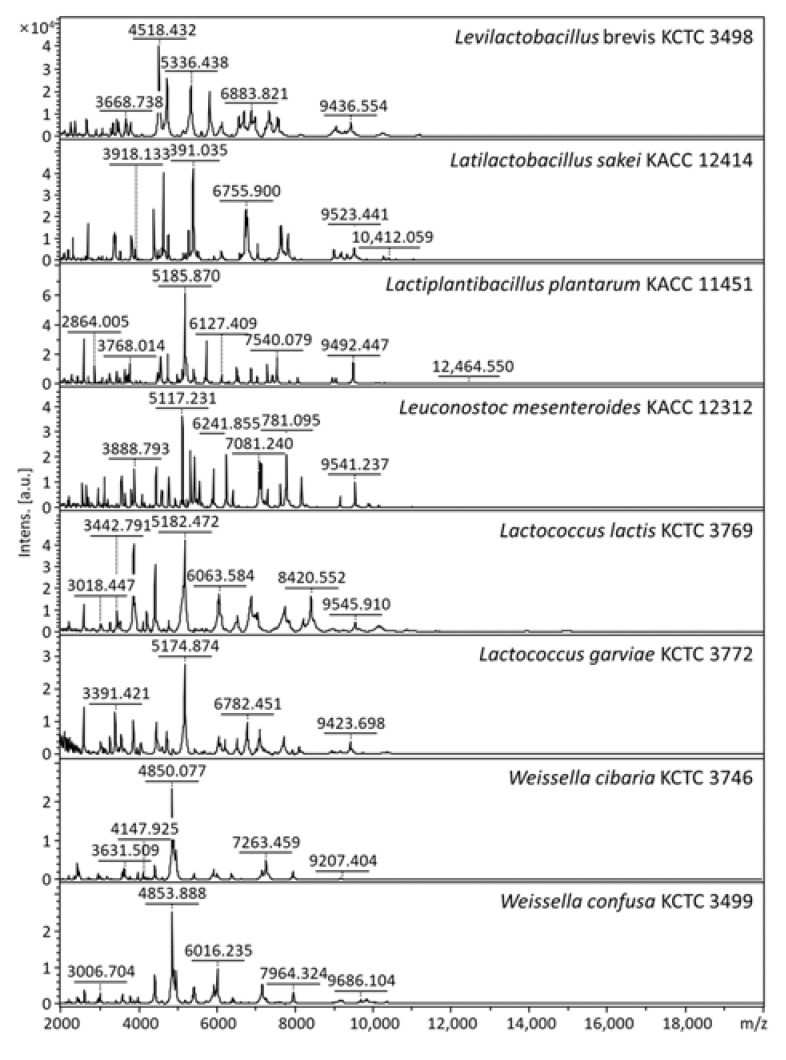
MALDI-TOF MS spectra of LAB mainly involved in kimchi fermentation; *m/z*, mass-to-charge ratio; a.u., arbitrary units.

**Figure 5 foods-10-01068-f005:**
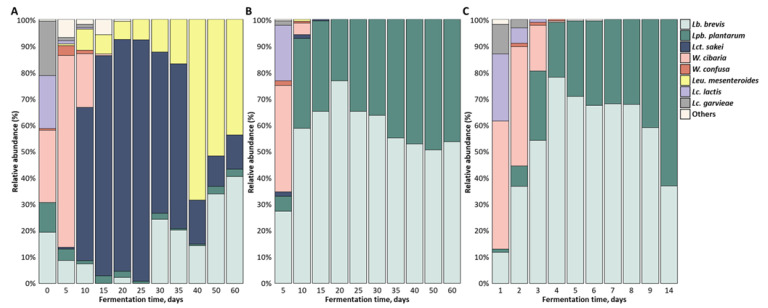
Changes in the microbial communities during the fermentation at (**A**) 4 °C, (**B**) 10 °C, and (**C**) 23 °C. The others indicate species with a prevalence of <0.4%, including *Lc*. *raffinolactis*, *Leu*. *lactis*, *P*. *ethanolidurans*, *E*. *sulfureus*, *E*. *cloacae*, *E*. *ludwigii*, *E*. *aerogenes*, and *S*. *simulans*.

**Figure 6 foods-10-01068-f006:**
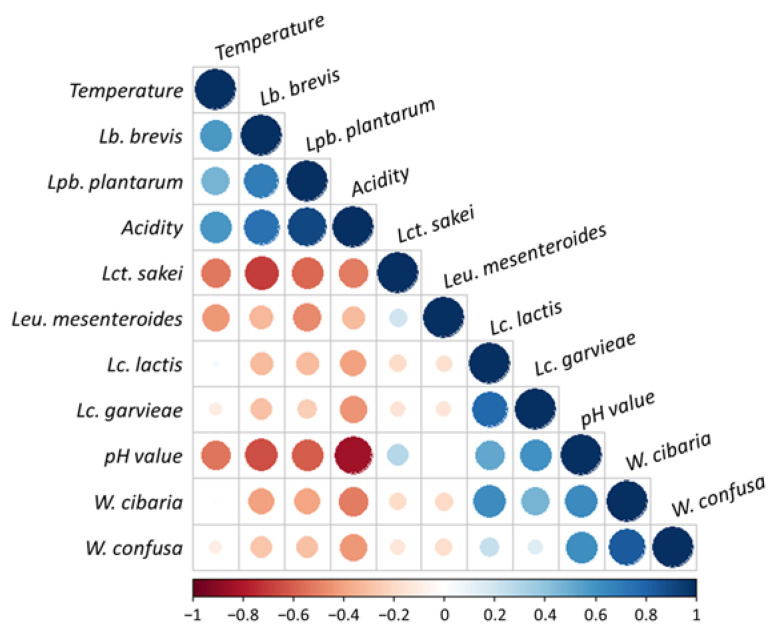
Pearson correlation matrix calculated from the relative abundance of major bacterial and environmental factors, such as pH, acidity, and fermented temperatures. The red and blue colors correspond to negative and positive correlations, respectively. The color intensity and circle size are proportional to the correlation coefficient. The color bar indicates the corresponding color and correlation coefficient.

**Table 1 foods-10-01068-t001:** Comparison of log score values obtained from fermented kimchi samples of microbials, based on MALDI-TOF in-house database.

Fermentation Temperature (No. of Isolates)	No. of Isolates with Results ^1^		
	≥2.000	1.700–1.999	≤1.699
4 °C (1733)	1466	267	0
10 °C (1733)	1490	243	0
23 °C (1738)	1511	227	0
Total isolates (5204)	4467	737	0

^1^ The meanings of log scores were as follows: ≥2.000, probable species; 1.700–1.999, probable genus; ≤1.699, non-reliable identification.

## Data Availability

The data presented in this study are available on request from the corresponding author.

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
