# Peer review of "Analysis of Cultivable Microbial Community during Kimchi Fermentation Using MALDI-TOF MS"

_foods, 2021, doi:10.3390/foods10051068_

Round 1

Reviewer 1 Report

The manuscript is well developed, the work is innovative, the objectives are clear, and the work well done. However, I suggest only a few changes, as below:

In the figures 2 and 4 should be changed axis title (X) (fermentation time, days)

Lines 186-187: it's not very clear the meaning, please rewrite the sentence (an alternative …to?)

Line 227: what does mean coeffect?

Line 259: it's not very clear the meaning, please rewrite the sentence

Author Response

The manuscript is well developed, the work is innovative, the objectives are clear, and the work well done. However, I suggest only a few changes, as below:

1. In the figures 2 and 4 should be changed axis title (X) (fermentation time, days)

Response: As you recommended, we changed the x-axis title in figures 2, 3, and 5.

2. Lines 186-187: it's not very clear the meaning, please rewrite the sentence (an alternative …to?)

Response: As you recommended, we revised the sentence in lines 193-194 as follows:

Lines 193-194: so, the ribosomal proteins can be used as an alternative to the 16S rRNA gene.

3. Line 227: what does mean coeffect?

Response: We changed Pearson coeffect to the Pearson correlation coefficient. As you recommended, we revised the sentence in lines 233, 250, 256, 257, 259, and 268 as follows:

Lines 233, 250, 256, 257, 259, and 268: Pearson coefficient

4. Line 259: it's not very clear the meaning, please rewrite the sentence

Response: As you recommended, we revised the sentence in lines 266-269 as follows:

Lines 266-269: Correlation between the abundance of species and the fermented temperature was analyzed, and the result indicated that Leu. mesenteroides was species having a negative correlation with fermented temperature (Pearson coefficient r = −0.432, p = 0.015).

Reviewer 2 Report

  1. Elements of scientific novelty should be presented in a detailed and convincing manner (in the last paragraph of the Introduction). In addition, it should also be briefly described in the Abstract.
  2. The current importance of the field should be clearly given in the Introduction.
  3. I suggest that a diagram (scheme) presenting the used synthesis method as well as analytical procedures used in the study should be added to Methods sub-section.
  4. In my opinion, as I see the green potential of the developed procedure, the green character should be evaluated. I recommend here, the GAPI index, as very easy tool. In the website, the tool which allow to create the GAPI pictogram is also available.
  5. Validation parameters characterized analytical procedure should be introduced.
  6. Future direction in this field should be recommended.

Author Response

1. Elements of scientific novelty should be presented in a detailed and convincing manner (in the last paragraph of the Introduction). In addition, it should also be briefly described in the Abstract.

Response: As you recommended, we added the sentence in lines 19-21 and 68-71 as follows:

Lines 19-21: This study shows as a novelty that MALDI-TOF MS is a robust and economically affordable method for investigating viable microbial community in kimchi.

Lines 68-71: The novelty of this study is to investigate the cultivable microbial community in kimchi. The previous studies conducted in this field have only focused on the investigation of microbial communities based on DNA, in which both live and dead microbial cells are detected [1,5,13,25].

2. The current importance of the field should be clearly given in the Introduction.

Response: As you recommended, we revised the sentence in lines 60-64 and 71-73 as follows:

Lines 60-64: In particular, the role of microorganisms during the kimchi fermentation is species-specific, and therefore it is critical to accurately analyze the microbial community at the species level. However, the previous studies conducted in this field have only focused on the high-throughput sequencing method, so that microbial community has been identified only at the genus level [1,5,13,25].

Lines 71-73: The DNA-based identification methods tend to overestimate bacterial abundance in a sample and fail to reflect the presence of microbial subpopulations with different viability states [28,29].

3. I suggest that a diagram (scheme) presenting the used synthesis method as well as analytical procedures used in the study should be added to Methods sub-section.

Response: As you recommended, we added a diagram presenting the used synthesis method and analytical procedures used in this study in lines 79 and 90-92 as follows:

Line 79: The overall procedure for this study is shown in Figure 1.

Lines 90-92: Diagram for analyzing the cultivable microbial community in kimchi. The strategy to identify microbial community involves: (1) sampling, (2) measurement of and viable cell counts, pH, and acidity, (3) isolation of kimchi isolates, and (4) identification of kimchi isolates.

We newly added a diagram for analytical procedures used in this study in Figure 1.

4. In my opinion, as I see the green potential of the developed procedure, the green character should be evaluated. I recommend here, the GAPI index, as very easy tool. In the website, the tool which allow to create the GAPI pictogram is also available.

Response: As you recommended, we evaluated the green character for the proposed procedure and added the sentence in lines 280-293 and 308-310 as follows:

Line 280: 3.5. Greenness assessment

Lines 281-293: The green analytical procedure index (GAPI) is a tool to evaluate the green character and the environmental impact of the entire analytical methodology, from the collection of samples to the final determination [39]. Evaluation of greenness of the proposed MALDI-TOF MS method was performed according to the GAPI tool [39]. Figure S1 demonstrates the GAPI pictograms organized according to data collected in Table S1. GAPI uses five pentagrams to assess environmental impacts at each step of the methodology with three color codes; green, yellow, and red representing low, medium, and high effects on the environment, respectively [40]. According to Figure S1, the GAPI pictogram for the proposed method has no red code with almost green zones except for the four yellow codes (middle impact), which correspond mainly to the sample extraction and safety of the reagents used. Therefore, the result from the pictogram indicates that the proposed method has eco-friendly due to the low consumption of reagents and solvents and small waste production in MALDI-TOF MS.

Lines 308-310: Figure S1: Greenness assessment profile for proposed MALDI-TOF MS method for investigating the microbial community in kimchi by GAPI tool

We newly added a green profile for the proposed procedure in Figure S1.

5. Validation parameters characterized analytical procedure should be introduced.

Response: As you recommended, we evaluated the analytical procedure according to 15 aspects of the green analytical procedure index parameters and added the result to Table S1.

Lines 310-311: Table S1: The green analytical procedure index of the proposed MALDI-TOF MS method.

We newly added validation parameters for the proposed procedure in Table S1.

6. Future direction in this field should be recommended.

Response: As you recommended, we added the sentence in lines 305-307 as follows:

Lines 305-307: The future direction is recommended to gain a deeper understanding of the microbial population by investigating the non-viable microbial communities in kimchi fermentation

Round 2

Reviewer 2 Report

I accept the current version of the manuscript